# Fluctuation-Dissipation Theorems for Multiphase Flow in Porous Media

**DOI:** 10.3390/e24010046

**Published:** 2021-12-27

**Authors:** Dick Bedeaux, Signe Kjelstrup

**Affiliations:** PoreLab, Department of Chemistry, Norwegian University of Science and Technology, NO-7491 Trondheim, Norway; signe.kjelstrup@ntnu.no

**Keywords:** fluctuation-dissipation theorems, nonequilibrium thermodynamics, flux-force relations, porous media

## Abstract

A thermodynamic description of porous media must handle the size- and shape-dependence of media properties, in particular on the nano-scale. Such dependencies are typically due to the presence of immiscible phases, contact areas and contact lines. We propose a way to obtain average densities suitable for integration on the course-grained scale, by applying Hill’s thermodynamics of small systems to the subsystems of the medium. We argue that the average densities of the porous medium, when defined in a proper way, obey the Gibbs equation. All contributions are additive or weakly coupled. From the Gibbs equation and the balance equations, we then derive the entropy production in the standard way, for transport of multi-phase fluids in a non-deformable, porous medium exposed to differences in boundary pressures, temperatures, and chemical potentials. Linear relations between thermodynamic fluxes and forces follow for the control volume. Fluctuation-dissipation theorems are formulated for the first time, for the fluctuating contributions to fluxes in the porous medium. These give an added possibility for determination of the Onsager conductivity matrix for transport through porous media. Practical possibilities are discussed.

## 1. Introduction

Porous media represent a vast and important class of systems; present for instance in biology, geology and in technological applications. Our interest is to describe transport in porous media. Examples are transport of nanoparticles in cancerous tissue, migrations in ground water reservoirs, or heterogeneous catalysis. A description of porous media transport must reflect on the macro-scale what goes on at the smaller scales. The choice of variables to be coarse-grained is therefore central. For porous media, it has been difficult to find good bottom-up descriptions of the complex, heterogeneous structure of the medium; as it contains a mixture of solids and often immiscible fluids, wetting or non-wetting, microporous and nanoporous. Methods that describe fluids confined to nanopores are also scarce, despite the documented large impact of confinement [1].

Porous media may be exposed to several external fields. Consider the case of a porous reservoir rock. On the macro-scale, we may have a variation in pressure in addition to the gravitational field. A gradient in composition may also be present. In the fractured carbonaceous Ekofisk oil field, a geothermal gradient was found to be relevant [2]. The interplay of several driving forces is then central. A suitable thermodynamic theory of transport must be able to describe such interplay and the energy dissipation involved (the entropy production). The fluctuations in each flux is a characteristic signature of the medium, that remains to be explored.

Thermodynamic theories of transport in homogeneous and heterogeneous fluids and solids have been studied over most of the last century and are well documented in nonequilibrium thermodynamics, see e.g., [3,4,5]. Furthermore, hydrodynamic fluctuations that are superimposed on the average flow behavior are well studied for bulk fluids [6]. In transport processes in porous media, there are co-operative effects which extend over multiple pores which occur on larger time scales. Not only viscous forces, but also capillarity acts in non-local ways [7,8,9,10,11,12]. Winkler et al. [7] simulated athermal flows in networks and reported relaxation times varying from 10${}^{-3}$ to 0.1 s for viscous flows. By including coalescence dynamics and interface phenomena on the pore scale, we may extend the scale of Winkler et al. at both ends; down to 10${}^{-5}$ s and up hours [9]. Fluid configuration shifts take up to minutes or more [8].

The attempts to characterize fluctuations which are typical to porous media, have so far concentrated on capillary pressure fluctuations [10,11]. Schlüter et al. [10] studied pressure relaxation in glass beads packs, using fast synchroton-based X-ray tomography, and found a slow pressure relaxation regime, of 1–4 h, preceded by a fast regime of seconds.

In Kubo’s formulation of the fluctuation dissipation theorem [13], the central issue is the fluctuations in addition to the slowly varying average flux. These fluctuations, not the pressure fluctuations, are the focus of the present work. Relevant time scales for the fluctuations of interest are therefore the microscopic time scales of the random contributions to the heat and mass fluxes. The averages of random contributions are zero, as they do not contribute to net flow. The fluctuations are of a thermal (e.g., viscous) nature and will characterize the Onsager coefficients in flux-force matrix. In particular, the fluctuations can be used to determine coupling coefficients, which are otherwise not easily available. Although there are random molecular fluctuations, there is also a collective movement that produces a systematic result, proportional to the velocity of the particles. Phenomena that are well separated on the time scale can be regarded as not coupled and dealt with separately. In order for various phenomena to interact or couple, they need to relax on (approximately) the same time scale.

Several attempts have been made to describe multiphase flow of this kind [14,15,16,17,18], all in essence aimed at a formulation in terms of average properties, that can describe measurements or simulations. Important general schemes using nonequilibrium thermodynamics were laid down by Hassanizadeh and Gray [16,17]. Helmig gave a pedagogical overview [19] of coarse graining methods. It was soon realized that a good definition of the representative elementary volume (REV) of the porous medium was crucial [14,16]. The REV must represent all porous medium microstates. But how many and which variables constitute a basis set? The coarse-graining technique is essential for the next steps in the derivation.

This work aims to define a set of REV variables, for which we can use the Gibbs equation on the REV-level, and therefore derive the entropy production for the REV. We will then be able to present fluctuation-dissipation theorems for the porous medium, which can serve as the Green–Kubo relations do in homogeneous systems to determine transport coefficients. Like Hassanizadeh and Gray [16,17] we shall use the entropy production as the governing property for transport phenomena of interest. But, rather than dealing with the total entropy production as a sum of contributions from the single phases, we shall define the total entropy production directly from the much smaller basis set of a coarse-grained variables. An attempt along the same lines was made [20] for microporous media. The entropy production for the REV will lead us directly to the thermodynamic flux-force relations, and define a basis for the fluctuation-dissipation theorems. Such theorems are, as far as we know, formulated here for the first time for transport through porous media. The consequences of this formulation are the same as for homogeneous fluids. The fluctuation-dissipation theorems provide an alternative route to obtain transport coefficients, including coupling coefficients, by simulations and experiments.

We propose that the solution to the problems described above can be resolved by Hill’s systematic method [21]. This gives a new method to define the REV variables. Hill’s thermodynamics is useful not only on the nanometer scale, but whenever the structure of a system calls for inclusion of size and shape effects. This is done by the introduction of the so-called replica energy, which includes all small system effects [22] and increases the number of variables by one. This is the first time this is done in a systematic manner for a set of variables of a nanoporous medium. Microporous pressure effects were discussed before [23,24].

The environment plays a crucial role in Hill’s theory, since the environment will control the REV variables. During up-scaling, the chemical potentials and the temperature are then defined by the outside reservoir. With the chemical potentials, temperature and volume being controlled, the grand potential is the relevant thermodynamic property, and the grand potential defines the replica energy, as defined by Hill [21,22].

The paper is outlined as follows. After a definition of the REV-variables (Section 2 and Section 3), we express the grand potential in terms of the grand partition function in Section 4. In Section 5, we discuss the meaning of equilibrium in a REV, including the validity of Gibbs equation for nanoporous media. This is the crucial step for the derivations that follow. With the REV and its Gibbs equation in place, we can find the entropy production (Section 6) and the constitutive equations (Section 7) in the standard way. This gives the corresponding fluctuation-dissipation theorems. The work is expanding on the analysis of nano- and microporous media [20] by adding the fluctuating contributions to the fluxes, and providing the corresponding fluctuation-dissipation theorems. The transport of two-phase flow of immiscible fluids is used in the last Sections as example to illustrate the equations [13,25,26]. We discuss how fluctuation-dissipation theorems provide direct routes to the Onsager coefficients of the system, and therefore to the more common permeabilities. The theorems offer an underused opportunity for determination of transport coefficients.

## 2. Porosity, Saturation, Surface Area and Contact Line Length

The definitions of variables particular to porous media in this section follow common procedures [19]. Consider a solid matrix of constant porosity ϕ [20]. The pore diameter can vary from nanometer to micrometer. Our system is homogeneous in the sense that porosity, saturation, surface areas and contact line lengths, on the average, are the same everywhere. The pores are filled with two immiscible fluid phases. We indicate the most wetting fluid with *w* and the least wetting fluid with *n*. We refer to them simply as the wetting and the non-wetting phases. The solid matrix is labeled *r*. Properties may later depend on the time, but this will not be indicated explicitly in the equations.

The present system is filled with three immiscible phases, so the chemical constituents are synonymous with a phase. The state of the REV can be characterized, among other variables, by the volumes of each fluid phase, $V^{n,\mathrm{REV}}$, $V^{w,\mathrm{REV}}$, and of the solid phase $V^{r,\mathrm{REV}}$. The total volume of the pores is
(1)$$V^{p,\mathrm{REV}}\equiv V^{n,\mathrm{REV}}+V^{w,\mathrm{REV}}$$
while the volume of the REV is
(2)$$V^{\mathrm{REV}}\equiv V^{r,\mathrm{REV}}+V^{p,\mathrm{REV}}+V^{nwr,\mathrm{REV}}$$

Superscript REV is used to indicate a property of the REV. The last term is the excess volume of the three-phase contact lines. While the excess volume of the surfaces is zero by definition, this is not the case for the three-phase contact lines. The reason is that the dividing surfaces may cross each other at three lines which have a slightly different location. The corresponding excess volume is in general very small, however. The volume of the REV is a so-called *additive variable.*

The porosity, ϕ, and the degree of saturation, or saturation as it is simply called, ${\tilde{S}}^{n}$ (and ${\tilde{S}}^{w}$) are given by
(3)$$\phi \equiv \frac{V^{p,\mathrm{REV}}}{V^{\mathrm{REV}}},\;\;\;\;{\tilde{S}}^{n}\equiv \frac{V^{n,\mathrm{REV}}}{V^{p,\mathrm{REV}}},\;\;\;{\tilde{S}}^{w}\equiv \frac{V^{w,\mathrm{REV}}}{V^{p,\mathrm{REV}}}$$

The porosity and the saturations are intensive variables. In our choice of REV, they do not depend on the size of the REV. They have therefore no superscript. It follows from the definitions that
(4)$${\tilde{S}}^{n}+{\tilde{S}}^{w}=1$$

In addition to the volumes of the bulk phases, there are interfacial areas between the phases in the REV: $\Omega^{nw,\mathrm{REV}},\;\Omega^{nr,\mathrm{REV}},\;\Omega^{wr,\mathrm{REV}}$. The total surface area of the solid phase is given by
(5)$$\Omega^{p,\mathrm{REV}}=\Omega^{nr,\mathrm{REV}}+\Omega^{wr,\mathrm{REV}}$$

The total length of the three phase contact lines is $\Lambda^{nwr,\mathrm{REV}}$. The surface areas and the contact line lengths are additive. This means that a REV of a double size has double the surface area of various types and double the line length. Two REVs have properties with double the value of one. So far our definitions are standard.

## 3. Thermodynamic Properties of the REV

We proceed to define the thermodynamic properties of a REV of volume $V^{\mathrm{REV}}$ following [20]. The proposal to use Minkowski functionals is similar [27,28]. We will take the REV large in the Hill sense. This implies that on the REV level, the additive variables of a REV with double the size are twice the value in the single REV. There is no dependence of the densities on the size of the REV. Additivity inside the REV is explained below. The additivity is given a statistical basis in Section 4.

### 3.1. Additive Variables

In addition to the volume, there are other additive REV variables, most prominently the masses. The mass of components *n*, *w*, *r* in the REV is the sum of bulk, excess interfacial, and excess line masses
(6)$$\begin{array}{lll} M_{n}^{\mathrm{REV}} & = & M_{n}^{n,\mathrm{REV}}+M_{n}^{nr,\mathrm{REV}}+M_{n}^{nw,\mathrm{REV}}+M_{n}^{nwr,\mathrm{REV}} \end{array}$$
(7)$$\begin{array}{lll} M_{w}^{\mathrm{REV}} & = & M_{w}^{w,\mathrm{REV}}+M_{w}^{wr,\mathrm{REV}}+M_{w}^{nwr,\mathrm{REV}} \end{array}$$
(8)$$\begin{array}{lll} M_{r}^{\mathrm{REV}} & = & M_{r}^{r,\mathrm{REV}} \end{array}$$

We have chosen to use the equimolar (or equimass) surface of the rock as the fluid–rock dividing surface and the wetting fluid dividing surface as the fluid–fluid dividing surface. Furthermore, we positioned the contact lines such that $M_{r}^{nwr,\mathrm{REV}}=0$. The total mass of each component in the REV is *independent* of our choice of the location of the dividing surfaces and contact line. When we use the various mass densities Equations (6)–(8) become
(9)$$\begin{array}{ccc} \rho_{n}V^{\mathrm{REV}} & = & \rho_{n}^{n}V^{n,\mathrm{REV}}+\rho_{n}^{nr}\Omega^{nr,\mathrm{REV}}+\rho_{n}^{nw}\Omega^{nw,\mathrm{REV}}+\rho_{n}^{nwr}\Lambda^{nwr,\mathrm{REV}} \end{array}$$
(10)$$\begin{array}{ccc} \rho_{w}V^{\mathrm{REV}} & = & \rho_{w}^{w}V^{w,\mathrm{REV}}+\rho_{w}^{wr}\Omega^{wr,\mathrm{REV}}+\rho_{w}^{nwr}\Lambda^{nwr,\mathrm{REV}} \end{array}$$
(11)$$\begin{array}{ccc} \rho_{r}V^{\mathrm{REV}} & = & \rho_{r}^{r}V^{r,\mathrm{REV}} \end{array}$$

The mass densities have dimensions kg·m${}^{-3}$ for the bulk phases, kg·m${}^{-2}$ for the surfaces and kg·m${}^{-1}$ for the contact line. The masses of the components are, like the volume, additive variables.

All densities refer to the REV. The REV is chosen such that if we increase the size of the REV, by for instance doubling its size, all variables with a superscript REV will double. (The variables are additive.) But this is not the case for the densities. They remain the same, independent of the size of the REV. This is in agreement with the early REV definitions [14]. All the densities of the bulk phases, surfaces and contact lines remain the same. Superscript REV is therefore not used for the densities.

The entropy of the REV is the sum of bulk, excess interfacial, excess line entropies:(12)$$\begin{array}{ccc} S^{\mathrm{REV}} & = & S^{\mathrm{mat}}+S^{\mathrm{conf}}=S^{n,\mathrm{REV}}+S^{w,\mathrm{REV}}+S^{r,\mathrm{REV}}+S^{nr,\mathrm{REV}}\\ & & +S^{wr,\mathrm{REV}}+S^{nw,\mathrm{REV}}+S^{nwr,\mathrm{REV}}+S^{\mathrm{conf}} \end{array}$$

The sum of bulk, excess interfacial, and excess line entropies is the *material contribution*, $S^{\mathrm{mat}}$, to the entropy. There are many ways to distribute the phases, and the origin of the *configurational contribution* comes from the many configurations that have the same component volumes, surface areas and contact line lengths in the REV. The integral of the logarithm of the corresponding probability distribution times $k_{B}$ gives the configurational contribution to the entropy. Both the material and the configurational contributions are additive. By introducing the entropy densities, the equation becomes
(13)$$\begin{array}{ccc} sV^{\mathrm{REV}} & = & \left(s^{\mathrm{mat}}+s^{\mathrm{conf}}\right)V^{\mathrm{REV}}=s^{n}V^{n,\mathrm{REV}}+s^{w}V^{w,\mathrm{REV}}+s^{r}V^{r,\mathrm{REV}}+s^{nr}\Omega^{nr,\mathrm{REV}}\\ & & +s^{wr}\Omega^{wr,\mathrm{REV}}+s^{nw}\Omega^{nw,\mathrm{REV}}+s^{nwr}\Lambda^{nwr,\mathrm{REV}}+s^{\mathrm{conf}}V^{\mathrm{REV}} \end{array}$$

The dimensions are J·K${}^{-1}$·m${}^{-3}$ for the bulk and configurational terms, J·K${}^{-1}$·m${}^{-2}$ for the excess interfacial entropy density, and J·K${}^{-1}$·m${}^{-1}$ for the excess line entropy density. For all variables relations similar to Equations (12) and (13) apply.

The internal energy of the REV is the sum of bulk, excess interfacial, excess line internal energies and a configurational contribution:(14)$$\begin{array}{ccc} U^{\mathrm{REV}} & = & U^{n,\mathrm{REV}}+U^{w,\mathrm{REV}}+U^{r,\mathrm{REV}}+U^{nr,\mathrm{REV}}+U^{wr,\mathrm{REV}}\\ & & +U^{nw,\mathrm{REV}}+U^{nwr,\mathrm{REV}}+U^{\mathrm{conf}} \end{array}$$

With the internal energy densities this equation becomes
(15)$$\begin{array}{ccc} uV^{\mathrm{REV}} & = & \left(u^{\mathrm{mat}}+u^{\mathrm{conf}}\right)V^{\mathrm{REV}}=u^{n}V^{n,\mathrm{REV}}+u^{w}V^{w,\mathrm{REV}}+u^{r}V^{r,\mathrm{REV}}+u^{nr}\Omega^{nr,\mathrm{REV}}\\ & & +u^{wr}\Omega^{wr,\mathrm{REV}}+u^{nw}\Omega^{nw,\mathrm{REV}}+u^{nwr}\Lambda^{nwr,\mathrm{REV}}+u^{\mathrm{conf}}V^{\mathrm{REV}} \end{array}$$

Their dimensions are J·m${}^{-3}$ for the bulk and configurational terms, J·m${}^{-2}$ for the excess interfacial energy density and J·m${}^{-1}$ for the excess line internal energy density. The above variables ($u,s,\rho_{n},\rho_{w}$) constitute the basis set of thermodynamic variables that will enter the Gibbs equation for the REV.

### 3.2. Dealing with Smallness: The Grand Potential and the
Replica Energy

The REV in porous media are often open to the surroundings. The grand potential is then relevant for a description. This potential concerns an open system of fixed volume with controlled temperature and chemical potentials. The surrounding reservoir is used for control. The medium can have nano-sized pores, and the pore fluids are then no longer like bulk fluids. We shall deal with change in thermodynamic variables using the so-called Small System Method, based on Hill’s original idea [29]. In this method a small representative elementary volume, $V^{\mathrm{REV}}$, is put into a bigger box. The bigger box controls the temperature and the chemical potentials. The statistical mechanical description is given by the grand canonical ensemble (GC). The grand potential of the REV is equal to minus the logarithm of the grand canonical partition function times $k_{B}T$.

The partition function can be used to support the choice of variables with additive contributions (see the next section). The grand potential $X^{\mathrm{GC},\mathrm{REV}}$ has material as well as configurational contributions. We have, similar to Equation (12) for the entropy, the grand potential
(16)$$\begin{array}{ccc} X^{\mathrm{GC},\mathrm{REV}} & = & X^{\mathrm{mat}}+X^{\mathrm{conf}}=X^{n,\mathrm{GC},\mathrm{REV}}+X^{w,\mathrm{GC},\mathrm{REV}}+X^{r,\mathrm{GC},\mathrm{REV}}+X^{nr,\mathrm{GC},\mathrm{REV}}\\ & & +X^{wr,\mathrm{GC},\mathrm{REV}}+X^{nw,\mathrm{GC},\mathrm{REV}}+X^{nwr,\mathrm{GC},\mathrm{REV}}+X^{\mathrm{conf}} \end{array}$$

By introducing the densities of the grand potential, this equation becomes
(17)$$\begin{array}{c} x^{\mathrm{GC}}V^{\mathrm{REV}}=\left(x^{\mathrm{mat}}+x^{\mathrm{conf}}\right)V^{\mathrm{REV}}=x^{n,\mathrm{GC}}V^{n,\mathrm{REV}}+x^{w,\mathrm{GC}}V^{w,\mathrm{REV}}+x^{r,\mathrm{GC}}V^{r,\mathrm{REV}}\\ +x^{nr,\mathrm{GC}}\Omega^{nr,\mathrm{REV}}+x^{wr,\mathrm{GC}}\Omega^{wr,\mathrm{REV}}+x^{nw,\mathrm{GC}}\Omega^{nw,\mathrm{REV}}+x^{nwr,\mathrm{GC}}\Lambda^{nwr,\mathrm{REV}}+x^{\mathrm{conf}}V^{\mathrm{REV}} \end{array}$$

The densities have dimension J·m${}^{-3}$ for a bulk and the configurational, J·m${}^{-2}$ for an excess interfacial and J·m${}^{-1}$ for the excess line grand potential densities. The grand potential densities do not depend on the size of the REV. We therefore dropped the superscript REV for the densities.

From the method of Hill [21], we define variables particular to subsystems in the REV which are small in his sense. These are systems which have considerable surface or line energies. The variables that describe the effect of smallness, or the deviation in the thermodynamical variables from the large system limit are the *hat*-variables in the GC ensemble. The hat-pressure, surface tension and line tensions were given by [21,22,29]:(18)$$\begin{array}{ccc} \hat{p} & = & -x^{\mathrm{GC}},\;\;\;{\hat{p}}^{\mathrm{mat}}=x^{\mathrm{mat}},\;\;\;{\hat{p}}^{\mathrm{conf}}=x^{\mathrm{conf}}\\ {\hat{p}}^{n} & = & -x^{n,\mathrm{GC}},\;\;\;{\hat{p}}^{w}=-x^{w,\mathrm{GC}},\;\;\;{\hat{p}}^{r}=-x^{r,\mathrm{GC}}\\ {\hat{\gamma }}^{nr} & = & x^{nr,\mathrm{GC}},\;\;\;{\hat{\gamma }}^{wr}=x^{wr,\mathrm{GC}},\;\;\;{\hat{\gamma }}^{nw}=x^{nw,\mathrm{GC}};\;\;\;{\hat{\gamma }}^{nwr}=x^{nwr,\mathrm{GC}} \end{array}$$

These quantities depend on the size of the pores when we approach the nano-scale, and are therefore unequal to the corresponding variable without a hat. The art is to choose the REV to be so large that $\hat{p}$, ${\hat{p}}^{\mathrm{mat}}$ and ${\hat{p}}^{\mathrm{conf}}$ do not depend on $V^{\mathrm{REV}}$. This must apply also for the nanoporous medium. For the REV, the material and the configurational pressures this implies that
(19)$$\hat{p}=p,\;\;\;\;{\hat{p}}^{\mathrm{mat}}=p^{\mathrm{mat}},\;\;\;\;{\hat{p}}^{\mathrm{conf}}=p^{\mathrm{conf}}$$

When we substitute Equations (6) and (18) into Equation (17) we obtain for the REV grand potential
(20)$$\begin{array}{c} pV^{\mathrm{REV}}=\left(p^{\mathrm{mat}}+p^{\mathrm{conf}}\right)V^{\mathrm{REV}}={\hat{p}}^{n}V^{n,\mathrm{REV}}+{\hat{p}}^{w}V^{w,\mathrm{REV}}+{\hat{p}}^{r}V^{r,\mathrm{REV}}\\ -{\hat{\gamma }}^{nr}\Omega^{nr,\mathrm{REV}}-{\hat{\gamma }}^{wr}\Omega^{wr,\mathrm{REV}}-{\hat{\gamma }}^{nw}\Omega^{nw,\mathrm{REV}}-{\hat{\gamma }}^{nwr}\Lambda^{nwr,\mathrm{REV}}+p^{\mathrm{conf}}V^{\mathrm{REV}} \end{array}$$

We show below why the grand potential $X^{\mathrm{GC},\mathrm{REV}}=-pV^{\mathrm{REV}}$ and as a consequence *p* have configurational contributions, in the case of system control by $T,V^{\mathrm{REV}},\mu^{n},\mu^{w},\mu^{r}$. Additivity implies that the pressure of the REV contains a weighted mean of the volume pressures minus the surface and line tensions. In this sense it becomes an effective pressure. As the surface tensions can be large, this may lower the REV pressure considerably.

### 3.3. Some REV-Size Considerations

Equations like Equation (15) can be used to estimate the relative size of bulk-, surface- and line-contributions. When we divide by $V^{\mathrm{REV}}$, we obtain
(21)$$\begin{array}{ccc} u & = & u^{n}\frac{V^{n,\mathrm{REV}}}{V^{\mathrm{REV}}}+u^{w}\frac{V^{w,\mathrm{REV}}}{V^{\mathrm{REV}}}+u^{r}\frac{V^{r,\mathrm{REV}}}{V^{\mathrm{REV}}}+u^{nr}\frac{\Omega^{nr,\mathrm{REV}}}{V^{\mathrm{REV}}}\\ & & +u^{wr}\frac{\Omega^{wr,\mathrm{REV}}}{V^{\mathrm{REV}}}+u^{nw}\frac{\Omega^{nw,\mathrm{REV}}}{V^{\mathrm{REV}}}+u^{nwr}\frac{\Lambda^{nwr,\mathrm{REV}}}{V^{\mathrm{REV}}}+u^{\mathrm{conf}} \end{array}$$

The bulk contributions are proportional to $V^{n,\mathrm{REV}}/V^{\mathrm{REV}}$, $V^{w,\mathrm{REV}}/V^{\mathrm{REV}}$, $V^{r,\mathrm{REV}}/V^{\mathrm{REV}}$, respectively, and are therefore generally of the same order of magnitude as *u*. The configurational contribution can be compared to an energy of mixing, and may have the same order of magnitude as *u*. The surface contributions are proportional to $\Omega^{nr,\mathrm{REV}}/V^{\mathrm{REV}}$, $\Omega^{wr,\mathrm{REV}}/V^{\mathrm{REV}}$, $\Omega^{nw,\mathrm{REV}}/V^{\mathrm{REV}}$, respectively, and therefore are in general smaller than *u*. The line contribution is proportional to $\Lambda^{nwr,\mathrm{REV}}/V^{\mathrm{REV}}$, and is therefore in general small compared to the surface contributions and much smaller than bulk contributions.

These statements about relative sizes depend on the magnitude of the excess densities. The excess mass of a surface active material can be large. Similarly, the surface tension times of the area can give a sizable contribution to the internal energy.

The REV must be large compared to the single pore volume, to provide representative average values. It should not be larger than necessary, however. In a study of a 2-dimensional network [30], 20 × 20 links were large enough to document validity of the ergodic hypothesis. The ergodic hypothesis was documented by experiments in the Hele–Shaw cell [31]. These findings support the idea that a thermodynamic description can be found. When the porous material consisted of a face centric array of large solid spheres, we found that half a unit cell was large enough to provide a REV [23]. The more regular the system is, the smaller the REV. The pores, or the substructure of the REV, might well be small [23]. When the substructure consists of nanosized pores, the excess densities will depend on the curvature of the surfaces and of the three phase lines.

## 4. Additive Contributions to the Grand Potential

Consider a REV of volume, $V^{\mathrm{REV}}$, with the temperature, *T*, the chemical potentials, $\mu^{n}$, $\mu^{w}$, $\mu^{r}$, kept constant by the surroundings. Additivity of the extensive variables can also be used to motivate the use of the grand potential. We use the grand canonical ensemble to calculate the grand canonical partition function, $\Xi (T,V^{\mathrm{REV}},\mu^{n},\mu^{w},\mu^{r})$ of the REV. The grand potential is given in terms of this partition function by
(22)$$X^{\mathrm{GC},\mathrm{REV}}\left(T,V^{\mathrm{REV}},\mu^{n},\mu^{w},\mu^{r}\right)=-k_{B}T\ln\Xi \left(T,V^{\mathrm{REV}},\mu^{n},\mu^{w},\mu^{r}\right)$$
where $k_{B}$ is the Boltzmann constant. The statistical mechanical treatment of small REV subsystems follows the procedure normal to large systems. The full distribution in phase space is normalized.

The contributions to the REV grand potential are additive because the wetting and non-wetting fluids, the rock, the surfaces and the contact line are *weakly coupled*. Subsystems can be said to be weakly coupled when their interaction energy is small compared to the energies of the separate subsystems, so that $H=H^{n}+H^{w}+H^{r}+H^{nr}+H^{wr}+H^{nwr}$, where the symbol H is a Hamiltonian.

The probability distribution in phase space is then equal to a product of the probability distributions in the subsystems
(23)$$\begin{array}{ccc} & & exp\left(-H/k_{B}T\right)=exp\left(-H^{n}/k_{B}T\right)exp\left(-H^{w}/k_{B}T\right)exp\left(-H^{r}/k_{B}T\right)\\ & & exp\left(-H^{nr}/k_{B}T\right)exp\left(-H^{wr}/k_{B}T\right)exp\left(-H^{nw}/k_{B}T\right)exp\left(-H^{nwr}/k_{B}T\right) \end{array}$$
For a given distribution of the fluids *the coordinates of the subsystems are restricted to their volumes, surface areas or contact lines*. It then follows that the grand canonical partition function of the REV is the product of the partition functions of the subsystems. The origin of the configurational contributions is a probability distribution in phase space which also contains a factor that give the probability distribution of the fluids over all possible choices of subvolumes. By integrating over phase space, we also integrates over these distributions, which lead to an additional partition function. All of this together gives
(24)$$\begin{array}{ccc} & & \Xi \left(T,V^{\mathrm{REV}},\mu^{n},\mu^{w},\mu^{r}\right)=\Xi^{n}\left(T,V^{n,\mathrm{REV}},\mu^{n}\right)\Xi^{w}\left(T,V^{w,\mathrm{REV}},\mu^{w}\right)\Xi^{r}\left(T,V^{r,\mathrm{REV}},\mu^{r}\right)\\ & & \times \Xi^{nr}\left(T,\Omega^{nr,\mathrm{REV}},\mu^{n},\mu^{r}\right)\Xi^{wr}\left(T,\Omega^{wr,\mathrm{REV}},\mu^{w},\mu^{r}\right)\Xi^{nw}\left(T,\Omega^{nw,\mathrm{REV}},\mu^{n},\mu^{w}\right)\\ & & \times \Xi^{nwr}\left(T,\Lambda^{nwr,\mathrm{REV}},\mu^{n},\mu^{w},\mu^{r}\right)\Xi^{\mathrm{conf}}\left(T,V^{\mathrm{REV}}\right) \end{array}$$

As the masses of the components have no configurational contribution, it follows that $\Xi^{\mathrm{conf}}$ does not depend on the chemical potentials.

When we now take the logarithm of Equation (24), multiply with $-k_{B}T$, and identify the contributions of the subsystems with the corresponding contributions to the grand potential, we obtain Equation (16):(25)$$\begin{array}{ccc} & & X^{\mathrm{GC},\mathrm{REV}}\left(T,V^{\mathrm{REV}},\mu^{n},\mu^{w},\mu^{r}\right)=X^{n,\mathrm{GC},\mathrm{REV}}\left(T,V^{n,\mathrm{REV}},\mu^{n}\right)\\ & & +X^{w,\mathrm{GC},\mathrm{REV}}\left(T,V^{w,\mathrm{REV}},\mu^{w}\right)+X^{r,\mathrm{GC},\mathrm{REV}}\left(T,V^{r,\mathrm{REV}},\mu^{r}\right)\\ & & +X^{nr,\mathrm{GC},\mathrm{REV}}\left(T,\Omega^{nr,\mathrm{REV}},\mu^{n},\mu^{r}\right)+X^{wr,\mathrm{GC},\mathrm{REV}}\left(T,\Omega^{wr,\mathrm{REV}},\mu^{w},\mu^{r}\right)\\ & & +X^{nw,\mathrm{GC},\mathrm{REV}}\left(T,\Omega^{nw,\mathrm{REV}},\mu^{n},\mu^{w}\right)+X^{nwr,\mathrm{GC},\mathrm{REV}}\left(T,\Lambda^{nwr,\mathrm{REV}},\mu^{n},\mu^{w},\mu^{r}\right)\\ & & +X^{\mathrm{conf}}\left(T,V^{\mathrm{REV}}\right) \end{array}$$

We have indicated the variables that the various contributions depend on. In the present case, we used that the components in the different bulk phases are immiscible. If they would be miscible, all material contributions depend on all the chemical potentials. Equations (17)–(20) are a direct consequence of Equation (26).

The autonomous nature of the surfaces and three-phase contact lines is a direct consequence of the weak coupling between subsystems. Examples of weakly coupled systems can be found everywhere. The catalytic surface with its own temperature, the liquid vapor interface at steady state, are two examples [5].

The volume and the masses of the components have only material contributions. This is not a consequence of weak coupling, however. The analysis in the next section will show that the Gibbs energy has a material contribution only. All other thermodynamic energies, like the internal energy, the Helmholtz energy, and the enthalpy, have configurational contributions.

The relevant properties are those averaged over a minimum volume of the REV. A discussion of averaging procedures and in particular of the minimum size of the REV was given by Pingaro et al. [32].

## 5. Gibbs Equation the Meaning of Equilibrium

The REV that we have considered so far is always in equilibrium with its surroundings. Away from global equilibrium the neighboring REVs are not necessarily in equilibrium with each other. The assumption of local equilibrium [5,6] covers this situation. It implies thermodynamic equilibrium within all REVs, and that the variables of the REV satisfy the Gibbs equation:(26)$$dU^{\mathrm{GC},\mathrm{REV}}=TdS^{\mathrm{GC},\mathrm{REV}}-pdV^{\mathrm{REV}}+\mu^{n}dM^{n,\mathrm{GC},\mathrm{REV}}+\mu^{w}dM^{w,\mathrm{GC},\mathrm{REV}}+\mu^{r}dM^{r,\mathrm{GC},\mathrm{REV}}$$

The application of the Gibbs equation to porous media is not new [16,17]. It was, however, not written for the REV variables before. The assumption, that this can be done, is a new, but natural extension, supported by the arguments of weakly coupled subsystems, use of additive variables, and system control by the environment. Each (additive) variable is constructed as described in Section 3. With this construction, we now can proceed as normal, for the coarse-grained variables. The predictions that follow need be confirmed by experiments and/or simulations.

Given that the REV is large in Hill’s sense $U^{\mathrm{GC},\mathrm{REV}}$ is an Euler homogeneous function of the first order, leading to:(27)$$U^{\mathrm{GC},\mathrm{REV}}=TS^{\mathrm{GC},\mathrm{REV}}-pV^{\mathrm{REV}}+\mu^{n}M^{n,\mathrm{GC},\mathrm{REV}}+\mu^{w}M^{w,\mathrm{GC},\mathrm{REV}}+\mu^{r}M^{r,\mathrm{GC},\mathrm{REV}}$$

By introducing the densities, Equation (26) becomes
(28)$$du^{\mathrm{GC}}=Tds^{\mathrm{GC}}+\mu^{n}d\rho^{n,\mathrm{GC}}+\mu^{w}d\rho^{w,\mathrm{GC}}+\mu^{r}d\rho^{r,\mathrm{GC}}$$

The corresponding Gibbs–Duhem relation is
(29)$$dp=s^{\mathrm{GC}}dT+\rho^{n,\mathrm{GC}}d\mu^{n}+\rho^{w,\mathrm{GC}}d\mu^{w}+\rho^{r,\mathrm{GC}}d\mu^{r}$$

In the analysis we have argued that $V^{\mathrm{REV}}$, $M^{n,\mathrm{GC},\mathrm{REV}},\;M^{w,\mathrm{GC},\mathrm{REV}},\;M^{r,\mathrm{GC},\mathrm{REV}}$ are material contributions, while $X^{\mathrm{GC},\mathrm{REV}}$, *p*, and $S^{\mathrm{GC},\mathrm{REV}}$ also have configurational contributions. The internal energy $U^{\mathrm{GC},\mathrm{REV}}$, the Helmholtz energy $F^{\mathrm{GC},\mathrm{REV}}$, the enthalpy $H^{\mathrm{GC},\mathrm{REV}}$, and the Gibbs energy $G^{\mathrm{GC},\mathrm{REV}}$ of the REV are given by
(30)$$\begin{array}{ccc} U^{\mathrm{GC},\mathrm{REV}} & = & TS^{\mathrm{GC},\mathrm{REV}}-pV^{\mathrm{REV}}+\mu^{n}M^{n,\mathrm{GC},\mathrm{REV}}+\mu^{w}M^{w,\mathrm{GC},\mathrm{REV}}+\mu^{r}M^{r,\mathrm{GC},\mathrm{REV}}\\ F^{\mathrm{GC},\mathrm{REV}} & = & -pV^{\mathrm{REV}}+\mu^{n}M^{n,\mathrm{GC},\mathrm{REV}}+\mu^{w}M^{w,\mathrm{GC},\mathrm{REV}}+\mu^{r}M^{r,\mathrm{GC},\mathrm{REV}}\\ H^{\mathrm{GC},\mathrm{REV}} & = & TS^{\mathrm{GC},\mathrm{REV}}+\mu^{n}M^{n,\mathrm{GC},\mathrm{REV}}+\mu^{w}M^{w,\mathrm{GC},\mathrm{REV}}+\mu^{r}M^{r,\mathrm{GC},\mathrm{REV}}\\ G^{\mathrm{GC},\mathrm{REV}} & = & \mu^{n}M^{n,\mathrm{GC},\mathrm{REV}}+\mu^{w}M^{w,\mathrm{GC},\mathrm{REV}}+\mu^{r}M^{r,\mathrm{GC},\mathrm{REV}} \end{array}$$

It follows that $G^{\mathrm{GC},\mathrm{REV}}$ has material contributions only.

The coarse-grained variables used in this section, are those which will enter the thermodynamic description on the macro-level. They are relevant in equations of transport, like the Darcy equation, the Washburn equation, or equations that follow from a nonequilibrium thermodynamic expression, see below.

We need assume that the Gibbs equation is valid for the REV also when transport takes place. This is called the assumption of local equilibrium. Droplets can form at high flow rates, while ganglia may occur at low rates. There is a minimum size of the REV, for which Gibbs equation can be written [23]. When we assume that Gibbs equation applies, we implicitly assume that there exists a uniquely defined state. The existence of such a state was first postulated by Hansen and Ramstad [33]. Experimental evidence for the assumption was found by Erpelding [31].

## 6. Entropy Production in Porous Media

We have seen above how we can define coarse-grained variables that describe a REV on the macroscale. Explicit expressions were given in terms of additive contributions. We did this for a REV in equilibrium meaning that *T*, $V^{\mathrm{REV}}$, $\mu^{n}$, $\mu^{w}$, $\mu^{r}$ are well defined. The Gibbs Equation (28) can then be written on the local form, in terms of densities. The discussion below considers *n* components. From now on, we omit the superscript REV, as all variables refer to the REV. The time rate of change of the internal energy density is then from Gibbs equation
(31)$$\frac{\partial s}{\partial t}=\frac{1}{T}\frac{\partial u}{\partial t}-\frac{1}{T}\sum_{i=1}^{n}\mu_{i}\frac{\partial \rho_{i}}{\partial t}$$

Gradients in mass and energy densities produce changes in the variables on the macro-scale. These lead to transport of heat and mass. The aim is to find the equations that govern this transport across the REV. The REV chosen is large enough to be macroscopic. The analysis closely follows a similar analysis for microporous media [20]. The balance equations for masses and internal energy densities of a REV are
(32)$$\begin{array}{ccc} \frac{\partial \rho_{i}}{\partial t} & = & -\frac{\partial }{\partial x}J_{i}\\ \frac{\partial u}{\partial t} & = & -\frac{\partial }{\partial x}J_{u}=-\frac{\partial }{\partial x}\left[J_{q}^{{}^{\prime }}+\sum_{i=1}^{n}J_{i}H_{i}\right] \end{array}$$

The transport on the REV-scale is in the x−direction only. The mass fluxes, $J_{i}$, and the flux of internal energy, $J_{u}$, are all macro-scale fluxes. The internal energy flux is the sum of the measurable (or sensible) heat flux, $J_{q}^{{}^{\prime }}$ and the partial specific enthalpies (latent heat), $H_{i}$ (in J·kg${}^{-1}$) times the component fluxes, $J_{i}$, see [3,5,17] for further explanations. Component *r* (the porous medium) is not moving and is the convenient frame of reference for the fluxes. The balance equations for the masses and the internal energy densities have their usual form.

The balance equation for the entropy balance on the REV-scale is:(33)$$\frac{\partial s}{\partial t}=-\frac{\partial }{\partial x}J_{s}+\sigma$$

Here $J_{s}$ is the entropy flux. Furthermore, σ is the entropy productions. The entropy production is positive definite, σ≥0 (the second law of thermodynamics).

We can now proceed to derive expressions for σ in the standard way [3,5], by combining the balance equations with Gibbs’ equation from the previous section. We introduce the balance equations for mass and energy into Gibbs equation, see [3,5] for details. By comparing the result with the entropy balance, Equation (31), we identify first the entropy flux
(34)$$J_{s}=\frac{J_{u}-\sum_{i=1}^{n}\mu_{i}J_{i}}{T}=\frac{1}{T}J_{q}^{\prime }+\sum_{i=1}^{n}J_{i}S_{i}$$

At the same time, we find the entropy production. Depending on the choice of independent fluxes, we can formulate the entropy production in various equivalent ways:(35)$$\sigma =J_{u}\frac{\partial }{\partial x}\left(\frac{1}{T}\right)-\sum_{i=1}^{n}J_{i}\frac{\partial }{\partial x}\left(\frac{\mu_{i}}{T}\right)=J_{q}^{\prime }\frac{\partial }{\partial x}\left(\frac{1}{T}\right)-\frac{1}{T}\sum_{i=1}^{n}J_{i}\frac{\partial }{\partial x}\mu_{i,T}$$

The entropy production, σ, is here expressed either using the energy flux, $J_{u}$, as a variable, or using the measurable heat flux, $J_{q}^{\prime }$ as a variable. The final choice is motivated by practical wishes; what is measurable or computable. When we choose $J_{u}$ as flux, the conjugate force is ∂(1/T)/∂x, and the mass fluxes $J_{i}$ are driven by minus the gradient in the Planck potential, $\mu_{i}/T$. When, on the other hand, we choose $J_{q}^{\prime }$ as the flux with the conjugate force ∂(1/T)/∂x, the mass fluxes are driven by minus the gradient in the chemical potential at constant temperature, divided by this temperature. The entropy production defines the independent thermodynamic driving forces and their conjugate fluxes. We have given two possible choices to demonstrate the flexibility [3,5,20]. The last expression is preferred for analysis of experiments.

The expression for the entropy production derived here applies to the coarse-grained description given of the REV in the preceding sections. The assumption of weakly coupled subsystems must apply. This is the first analysis of porous media transport in terms of such coarse-grained variables. Experiments and/or simulations are needed for further developments.

## 7. Constitutive Equations

From the entropy production we obtain the following flux-force relations when we use $J_{u}$:(36)$$\begin{array}{ccc} J_{u} & = & \ell_{uu}\frac{\partial }{\partial x}\left(\frac{1}{T}\right)-\sum_{i=1}^{n}\ell_{ui}\frac{\partial }{\partial x}\left(\frac{\mu_{i}}{T}\right)\\ J_{j} & = & \ell_{ju}\frac{\partial }{\partial x}\left(\frac{1}{T}\right)-\sum_{i=1}^{n}\ell_{ji}\frac{\partial }{\partial x}\left(\frac{\mu_{i}}{T}\right) \end{array}$$

According to Onsager [34], the conductivity matrix is symmetric:(37)$$\ell_{ju}=\ell_{uj},\;\;\;\ell_{ji}=\ell_{ij}$$

This is so, independent of the mechanism of transport. The magnitude of the coefficients reflect, of course, the mechanism of transport, but can not be used to conclude on the mechanism in play.

When we use $J_{q}^{\prime }$, we obtain
(38)$$\begin{array}{ccc} J_{q}^{\prime } & = & L_{qq}\frac{\partial }{\partial x}\left(\frac{1}{T}\right)-\frac{1}{T}\sum_{i=1}^{n}L_{qi}\frac{\partial }{\partial x}\mu_{i,T}\\ J_{j} & = & L_{jq}\frac{\partial }{\partial x}\left(\frac{1}{T}\right)-\frac{1}{T}\sum_{i=1}^{n}L_{ji}\frac{\partial }{\partial x}\mu_{i,T} \end{array}$$

This conductivity matrix is also symmetric:(39)$$L_{jq}=L_{qj},\;\;\;L_{ji}=L_{ij}$$

By using the relation between the $J_{u}$ and $J_{q}^{\prime }$ we can express one conductivity matrix, say Equations (36), in terms of the other matrix (of Equation (38)), or vice versa. The symmetry can only be expected for properly constructed conjugate thermodynamic forces and fluxes, see for instance the discussion in [3,4,5]. The matrix of Fick diffusion coefficients, for instance, will not be symmetric for this reason.

## 8. Fluctuation-Dissipation Theorems

We have so far discussed how fluids flow through a porous medium, governed by the entropy production. All fluxes, densities, temperatures, pressures and chemical potentials were defined on the coarse-grained scale (not on the pore-scale), as explained above. The assumptions used in these Sections are considered to apply. The description is now extended to incorporate fluctuations. We do this by adding fluctuating contributions to the fluxes, which are indicated by subscript *R*. These fluctuating contributions are crucial in the formulation of fluctuation-dissipation theorems [13,25,26]. The contributions we refer to are the quickly changing molecular contributions. Their temporal and spatial correlations are therefore on a molecular scale. Course grained this gives δ-functions in space and time on the macroscopic space and time scales. Their strength is given by $2k_{B}$ times the Onsager conductivity matrix.

The properties of the fluctuations are the usual ones, as they appear also in the derivation of fluctuation-dissipation theorems [13,25,26]. We assume these to be valid also for flows in porous media. Again, the predictions will have to be validated. We are only aware of one study so far, see the Discussion section.

For the internal energy flux, we obtain
(40)$$\begin{array}{ccc} J_{u,tot} & = & J_{u}+J_{u,R}\\ J_{j,tot} & = & J_{j}+J_{j,R} \end{array}$$

When we use the measurable heat flux, we obtain
(41)$$\begin{array}{ccc} J_{q,tot}^{\prime } & = & J_{q}^{\prime }+J_{q,R}^{\prime }\\ J_{j,tot} & = & J_{j}+J_{j,R} \end{array}$$

All fluxes in these equation are local-valued fluxes, and not fluxes integrated over the cross section of the porous medium normal to the direction of flow. The averages of the random contributions to the fluxes are equal to zero
(42)$$\langle J_{u,R}\rangle =\langle J_{q,R}^{\prime }\rangle =\langle J_{j,R}\rangle =0$$

This implies that
(43)$$\begin{array}{ccc} \langle J_{u,tot}\rangle & = & \langle J_{u}\rangle \\ \langle J_{q,tot}^{\prime }\rangle & = & \langle J_{q}^{\prime }\rangle \\ \langle J_{j,tot}\rangle & = & \langle J_{j}\rangle \end{array}$$

On the REV time-scale, the random contributions are Gaussian white noise. Similarly, their spatial correlations are short range compared to the macroscale phenomena. Their second moments follow from the statistical mechanical description and satisfy the fluctuation-dissipation theorem [13,25,26]. The conditions of Onsager symmetry apply. In the case of the internal energy flux, this gives
(44)$$\begin{array}{ccc} \langle J_{u,R}\left(r,t\right)J_{u,R}\left(r^{\prime },t^{\prime }\right)\rangle & = & 2k_{B}\ell_{uu}\delta \left(r-r^{\prime }\right)\delta \left(t-t^{\prime }\right)\\ \langle J_{u,R}\left(r,t\right)J_{j,R}\left(r^{\prime },t^{\prime }\right)\rangle & = & \langle J_{j,R}\left(r,t\right)J_{u,R}\left(r^{\prime },t^{\prime }\right)\rangle =2k_{B}\ell_{uj}\delta \left(r-r^{\prime }\right)\delta \left(t-t^{\prime }\right)\\ \langle J_{i,R}\left(r,t\right)J_{j,R}\left(r^{\prime },t^{\prime }\right)\rangle & = & \langle J_{j,R}\left(r,t\right)J_{i,R}\left(r^{\prime },t^{\prime }\right)\rangle =2k_{B}l_{ij}\delta \left(r-r^{\prime }\right)\delta \left(t-t^{\prime }\right) \end{array}$$

The δ-functions reflect the short range nature of the spatial and temporal correlations.

In the case that the measurable heat flux is a variable, we have
(45)$$\begin{array}{ccc} \langle J_{q,R}^{\prime }\left(r,t\right)J_{q,R}^{\prime }\left(r^{\prime },t^{\prime }\right)\rangle & = & 2k_{B}L_{qq}\delta \left(r-r^{\prime }\right)\delta \left(t-t^{\prime }\right)\\ \langle J_{q,R}^{\prime }\left(r,t\right)J_{j,R}\left(r^{\prime },t^{\prime }\right)\rangle & = & \langle J_{j,R}\left(r,t\right)J_{q,R}^{\prime }\left(r^{\prime },t^{\prime }\right)\rangle =2k_{B}L_{qj}\delta \left(r-r^{\prime }\right)\delta \left(t-t^{\prime }\right)\\ \langle J_{i,R}\left(r,t\right)J_{j,R}\left(r^{\prime },t^{\prime }\right)\rangle & = & \langle J_{j,R}\left(r,t\right)J_{i,R}\left(r^{\prime },t^{\prime }\right)\rangle =2k_{B}L_{ij}\delta \left(r-r^{\prime }\right)\delta \left(t-t^{\prime }\right) \end{array}$$

The fluctuations, and therefore the correlation, of a flux pair are characteristic for the pair. The correlation of fluctuation in the mass flux $J_{r,R}$ with the internal energy flux, differ from the corresponding correlation with the measurable heat flux. Self-correlations give information of diagonal coefficients, while cross-correlations give information of the cross-coefficients. One set of fluctuation-dissipation relations can be derived from the other, and vice versa. The fluctuating flux–flux correlation functions decay fast compared to the macroscopic time scale [13,25,26].

The expressions apply for any condition, steady state or not, and do not refer to the nature of the flux-force relationships. The theorems apply, not only in equilibrium, but also away from equilibrium [3,6].

## 9. The Chemical Potential at Constant Temperature

The derivative of the chemical potential at constant temperature is needed in the driving forces in the previous section. For convenience we repeat its relation to the full chemical potential [3]. The differential of the full chemical potential is:(46)$$d\mu_{i}=-S_{i}dT+V_{i}dp+\sum_{j=1}^{n}{\left(\frac{\partial \mu_{i}}{\partial M_{j}}\right)}_{p,T,M_{i}}dM_{j}$$
where $S_{i},V_{i}$ and ${\left(\partial \mu_{i}/\partial M_{j}\right)}_{p,T,M_{i}}$ are partial specific quantities. The partial specific entropies and volumes are equal to:(47)$$S_{i}=-{\left(\frac{\partial \mu_{i}}{\partial T}\right)}_{p,M_{j}}\;\;\;,\;\;\;V_{i}={\left(\frac{\partial \mu_{i}}{\partial p}\right)}_{T,M_{j}}$$
and the last term of Equation (46) is denoted by
(48)$$d\mu_{i}^{c}\equiv \sum_{j=1}^{n}{\left(\frac{\partial \mu_{i}}{\partial M_{j}}\right)}_{p,T,M_{i}}dM_{j}$$

By reshuffling, we have the quantity of interest as the differential of the full chemical potential plus an entropic term;
(49)$$d\mu_{i,T}\equiv d\mu_{i}+S_{i}dT=V_{i}dp+d\mu_{i}^{c}$$

The Gibbs-Duhem’s equation is $0=SdT-Vdp+\sum_{j=1}^{n}\rho_{j}d\mu_{j}$. By introducing Equation (49) into this equation we obtain an equivalent expression, to be used below:(50)$$0=\sum_{j=1}^{n}\rho_{j}d\mu_{j}^{c}$$

## 10. Example: Immiscible Two-Phase Flow

### 10.1. The Entropy Production

Consider the case of two immiscible fluids, one more wetting (w) and one more non-wetting (n). The entropy production in Equation (35) gives
(51)$$\sigma =J_{q}^{\prime }\frac{\partial }{\partial x}\left(\frac{1}{T}\right)-J_{w}\frac{1}{T}\frac{\partial }{\partial x}\mu_{w,T}-J_{n}\frac{1}{T}\frac{\partial }{\partial x}\mu_{n,T}$$

The solid matrix is the frame of reference for transport, $J_{r}=0$ and does not contribute to the entropy production. The volume flux is frequently measured, and we wish to introduce this as new variable
(52)$$J_{V}=J_{n}V_{n}+J_{w}V_{w}$$

Here, $J_{V}$ has the dimension of a velocity (m·s${}^{-1}$), and the partial specific volumes have dimension m${}^{3}$·kg${}^{-1}$. The chemical potential of the solid matrix may not vary much if the composition of the solid is constant across the system. We assume that this is the case ($d\mu_{m}^{c}\approx 0$), and use Equation (50) to obtain
(53)$$0=\rho_{n}d\mu_{n}^{c}+\rho_{w}d\mu_{w}^{c}$$

The entropy production is invariant to the choice of variables. We can introduce the relations above and the explicit expression for $d\mu_{i,T}$ into Equation (51), and find:(54)$$\sigma =J_{q}^{\prime }\frac{\partial }{\partial x}\left(\frac{1}{T}\right)-J_{V}\frac{1}{T}\frac{\partial p}{\partial x}-J_{D}\frac{\rho_{w}}{T}\frac{\partial \mu_{w}^{c}}{\partial x}$$
where we used Equations (52) and (53) and the difference velocity $J_{D}$:(55)$$J_{D}=\frac{J_{w}}{\rho_{w}}-\frac{J_{n}}{\rho_{n}}$$

The difference velocity describes the relative movement of the two components within the porous matrix. In other words, it describes the ability of the medium to separate components. The main driving force for separation is the chemical driving force, related to the gradient of the saturation. The equation implies that also temperature and pressure gradients may play a role for the separation.

The entropy production has again three terms, one for each independent driving force. With a single fluid, the number of terms are two. The force conjugate to the heat flux is again the gradient of the inverse temperature. The entropy production, in the form of Equation (51) or Equation (54), dictate the constitutive equations of the system.

The total volume flows through the REV $Q_{n}\equiv AJ_{n}V_{n}$ and $Q_{w}\equiv AJ_{w}V_{w}$ were used by Hansen et al. [35]. Here A≡V/ℓ is the cross section, and *ℓ* is the length of the REV in the flow direction. Hansen et al. [35] assumed that the total volume flow was a Euler homogeneous function of first order in the fractional areas, $A^{n}\equiv V_{n}/\ell =\left(V_{n}/V_{p}\right)V_{p}/\ell ={\tilde{S}}_{n}A^{p}$ and $A^{w}\equiv V_{w}/\ell =\left(V_{w}/V_{p}\right)V_{p}/\ell ={\tilde{S}}_{w}A^{p}$. Here ${\tilde{S}}_{n}=V_{n}/V_{p}=A_{n}/A_{p}$ and ${\tilde{S}}_{w}=V_{w}/V_{p}=A_{w}/A_{p}$ represent the saturation of the non-wetting and wetting fluid, respectively. Furthermore, the porosity is given by $\phi =V_{p}/V=A_{p}/A$. We will also use the total measurable heat flow $Q_{q}\equiv AJ_{q}^{\prime }$. (We will not use the so-called seepage velocities [35], which were defined as $v_{n}\equiv Q_{n}/A^{n}$ and $v_{w}\equiv Q_{w}/A^{w}$.) Because of the REV being homogeneous in the y,z directions, the total flows are the integrals of the flows across the cross-sectional area of the REV normal to the flow direction. By introducing the total flows in the expression for the total entropy production, Equation (51), we obtain:(56)$$\Sigma \equiv A\sigma =Q_{q}\frac{\partial }{\partial x}\left(\frac{1}{T}\right)-Q_{w}\frac{1}{TV_{w}}\frac{\partial }{\partial x}\mu_{w,T}-Q_{n}\frac{1}{TV_{n}}\frac{\partial }{\partial x}\mu_{n,T}$$

Similarly, Equation (54) becomes
(57)$$\Sigma \equiv A\sigma =Q_{q}\frac{\partial }{\partial x}\left(\frac{1}{T}\right)-Q_{V}\frac{1}{T}\frac{\partial p}{\partial x}-Q_{D}\frac{\rho_{w}}{T}\frac{\partial \mu_{w}^{c}}{\partial x}$$
where $Q_{V}\equiv AJ_{V}$ and $Q_{D}\equiv AJ_{D}$. All the total fluxes as well as the corresponding thermodynamic forces only depend on *x* and *t* and not on *y* and *z*.

### 10.2. Constitutive Equations

The entropy production given by Equations (51) or Equation (54), offer three equivalent choices for constitutive equations. For the sake of completeness, we give all of them.

Equation (51) dictates the following flux-force relations
(58)$$\begin{array}{ccc} J_{q}^{\prime } & = & L_{qq}\frac{\partial }{\partial x}\left(\frac{1}{T}\right)-\frac{1}{T}L_{qw}\frac{\partial }{\partial x}\mu_{w,T}-\frac{1}{T}L_{qn}\frac{\partial }{\partial x}\mu_{n,T}\\ J_{w} & = & L_{wq}\frac{\partial }{\partial x}\left(\frac{1}{T}\right)-\frac{1}{T}L_{ww}\frac{\partial }{\partial x}\mu_{w,T}-\frac{1}{T}L_{wn}\frac{\partial }{\partial x}\mu_{n,T}\\ J_{n} & = & L_{nq}\frac{\partial }{\partial x}\left(\frac{1}{T}\right)-\frac{1}{T}L_{nw}\frac{\partial }{\partial x}\mu_{w,T}-\frac{1}{T}L_{nn}\frac{\partial }{\partial x}\mu_{n,T} \end{array}$$

These are the equations given in Equation (8) for the special case of two fluids. According to Onsager [34,36], the conductivity matrix is symmetric.

Equation (54) dictates the following flux-force relations
(59)$$\begin{array}{ccc} J_{q}^{\prime } & = & L_{qq}^{D}\frac{\partial }{\partial x}\left(\frac{1}{T}\right)-\frac{1}{T}L_{qV}^{D}\frac{\partial p}{\partial x}-\frac{1}{T}L_{qw}^{D}\frac{\partial \mu_{w}^{c}}{\partial x}\\ J_{V} & = & L_{Vq}^{D}\frac{\partial }{\partial x}\left(\frac{1}{T}\right)-\frac{1}{T}L_{VV}^{D}\frac{\partial p}{\partial x}-\frac{1}{T}L_{VD}^{D}\frac{\partial \mu_{w}^{c}}{\partial x}\\ J_{D} & = & L_{Dq}^{D}\frac{\partial }{\partial x}\left(\frac{1}{T}\right)-\frac{1}{T}L_{DV}^{D}\frac{\partial p}{\partial x}-\frac{1}{T}L_{DD}^{D}\frac{\partial \mu_{w}^{c}}{\partial x} \end{array}$$

Also this conductivity matrix is symmetric.

Equation (56) dictates the following flux-force relations
(60)$$\begin{array}{ccc} Q_{q} & = & L_{qq}^{P}\frac{\partial }{\partial x}\left(\frac{1}{T}\right)-\frac{L_{qw}^{P}}{TV_{w}}\frac{\partial }{\partial x}\mu_{w,T}-\frac{L_{qn}^{P}}{TV_{n}}\frac{\partial }{\partial x}\mu_{n,T}\\ Q_{w} & = & L_{wq}^{P}\frac{\partial }{\partial x}\left(\frac{1}{T}\right)-\frac{L_{ww}^{P}}{TV_{w}}\frac{\partial }{\partial x}\mu_{w,T}-\frac{L_{wn}^{P}}{TV_{n}}\frac{\partial }{\partial x}\mu_{n,T}\\ Q_{n} & = & L_{nq}^{P}\frac{\partial }{\partial x}\left(\frac{1}{T}\right)-\frac{L_{nw}^{P}}{TV_{w}}\frac{\partial }{\partial x}\mu_{w,T}-\frac{L_{nn}^{P}}{TV_{n}}\frac{\partial }{\partial x}\mu_{n,T} \end{array}$$
and Equation (57) gives
(61)$$\begin{array}{ccc} Q_{q} & = & L_{qq}^{A}\frac{\partial }{\partial x}\left(\frac{1}{T}\right)-\frac{1}{T}L_{qV}^{A}\frac{\partial p}{\partial x}-\frac{1}{T}L_{qw}^{A}\frac{\partial \mu_{w}^{c}}{\partial x}\\ Q_{V} & = & L_{Vq}^{A}\frac{\partial }{\partial x}\left(\frac{1}{T}\right)-\frac{1}{T}L_{VV}^{A}\frac{\partial p}{\partial x}-\frac{1}{T}L_{VD}^{A}\frac{\partial \mu_{w}^{c}}{\partial x}\\ Q_{D} & = & L_{Dq}^{A}\frac{\partial }{\partial x}\left(\frac{1}{T}\right)-\frac{1}{T}L_{DV}^{A}\frac{\partial p}{\partial x}-\frac{1}{T}L_{DD}^{A}\frac{\partial \mu_{w}^{c}}{\partial x} \end{array}$$

Also these conductivity matrices are symmetric. From the relations between the fluxes, we find relations between the conductivity matrices of Equations (59)–(62). In this way we find from Equations (59) and (61) that
(62)$$\begin{array}{ccc} L_{qq}^{P} & = & AL_{qq},\;\;\;L_{qw}^{P}=AV_{w}L_{qw},\;\;\;L_{qn}^{P}=AV_{n}L_{qn}\\ L_{ww}^{P} & = & AV_{w}^{2}L_{ww},\;\;\;L_{wn}^{P}=AV_{w}V_{n}L_{wn},\;\;\;L_{nn}^{P}=AV_{n}^{2}L_{nn} \end{array}$$
and from Equations (60) and (62) that
(63)$$L_{ij}^{A}=AL_{ij}^{D}\;\;\;\mathrm{for}\;\;\;i,j=q,V,D$$

### 10.3. Fluctuation-Dissipation Theorems for the Two-Fluid Mixture in a
Porous Medium

As we discussed above, there are fluctuations in the fluxes of the coarse-grained variables. Each flux $J_{q}^{\prime }$, $J_{w}$
$J_{n}$ can be given in terms of a mean value plus a fluctuating (random) contribution [13,25,26]:(64)$$\begin{array}{ccc} J_{q,tot}^{\prime } & = & J_{q}^{\prime }+J_{q,R}^{\prime }\\ J_{w,tot} & = & J_{w}+J_{w,R}\\ J_{n,tot} & = & J_{n}+J_{n,R} \end{array}$$

Again, the expressions apply for steady or non-steady states, independent of the the nature of the flux-force relationships. The theorems apply not only in equilibrium, but also away from equilibrium [3]. On the REV time scale, the random contributions are typically Gaussian white noise. Similarly, their spatial correlations are short-range compared to the macroscale phenomena. Their second moments satisfy the fluctuation-dissipation theorem
(65)$$\begin{array}{ccc} \langle J_{q,R}^{\prime }\left(r,t\right)J_{q,R}^{\prime }\left(r^{\prime },t^{\prime }\right)\rangle & = & 2k_{B}L_{qq}\delta \left(r-r^{\prime }\right)\delta \left(t-t^{\prime }\right)\\ \langle J_{q,R}^{\prime }\left(r,t\right)J_{j,R}\left(r^{\prime },t^{\prime }\right)\rangle & = & \langle J_{j,R}\left(r,t\right)J_{q,R}^{\prime }\left(r^{\prime },t^{\prime }\right)\rangle =2k_{B}L_{qj}\delta \left(r-r^{\prime }\right)\delta \left(t-t^{\prime }\right)\\ \langle J_{i,R}\left(r,t\right)J_{j,R}\left(r^{\prime },t^{\prime }\right)\rangle & = & \langle J_{j,R}\left(r,t\right)J_{i,R}\left(r^{\prime },t^{\prime }\right)\rangle =2k_{B}L_{ij}\delta \left(r-r^{\prime }\right)\delta \left(t-t^{\prime }\right) \end{array}$$
where i,j are either *w* or *n*.

Likewise, for $J_{q}^{\prime }$, $J_{V}$, $J_{D}$ we have
(66)$$\begin{array}{ccc} J_{q,tot}^{\prime } & = & J_{q}^{\prime }+J_{q,R}^{\prime }\\ J_{V,tot} & = & J_{V}+J_{V,R}\\ J_{D,tot} & = & J_{D}+J_{D,R} \end{array}$$

The averages of the fluctuating contributions are zero. Their second moments satisfy the fluctuation-dissipation theorem
(67)$$\begin{array}{ccc} \langle J_{q,R}^{\prime }\left(r,t\right)J_{q,R}^{\prime }\left(r^{\prime },t^{\prime }\right)\rangle & = & 2k_{B}L_{qq}^{D}\delta \left(r-r^{\prime }\right)\delta \left(t-t^{\prime }\right)\\ \langle J_{q,R}^{\prime }\left(r,t\right)J_{j,R}\left(r^{\prime },t^{\prime }\right)\rangle & = & \langle J_{j,R}\left(r,t\right)J_{q,R}^{\prime }\left(r^{\prime },t^{\prime }\right)\rangle =2k_{B}L_{qj}^{D}\delta \left(r-r^{\prime }\right)\delta \left(t-t^{\prime }\right)\\ \langle J_{i,R}\left(r,t\right)J_{j,R}\left(r^{\prime },t^{\prime }\right)\rangle & = & \langle J_{j,R}\left(r,t\right)J_{i,R}\left(r^{\prime },t^{\prime }\right)\rangle =2k_{B}L_{ij}^{D}\delta \left(r-r^{\prime }\right)\delta \left(t-t^{\prime }\right) \end{array}$$
where i,j are either *V* or *D*. For $Q_{q}$, $Q_{w}$, $Q_{n}$ we have
(68)$$Q_{i,tot}=Q_{i}+Q_{i,R}$$

The averages of the fluctuating contributions is zero. Their second moments satisfy the fluctuation-dissipation theorem
(69)$$\begin{array}{ccc} \langle Q_{i,R}\left(x,t\right)Q_{j,R}\left(x^{\prime },t^{\prime }\right)\rangle & = & \langle Q_{j,R}\left(x,t\right)Q_{i,R}\left(x^{\prime },t^{\prime }\right)\rangle =2k_{B}L_{ij}^{P}\delta \left(x-x^{\prime }\right)\delta \left(t-t^{\prime }\right) \end{array}$$

In Equations (68) and (69) i,j are either q,w or *n*. The fluctuation-dissipation relations can be derived from Equation (66) by integrating over the cross section.

For $Q_{q}$, $Q_{V}$, $Q_{D}$ we have
(70)$$Q_{i,tot}=Q_{i}+Q_{i,R}$$

The averages of the fluctuating contributions is zero. Their second moments satisfy the fluctuation-dissipation theorem
(71)$$\begin{array}{ccc} \langle Q_{i,R}\left(x,t\right)Q_{j,R}\left(x^{\prime },t^{\prime }\right)\rangle & = & \langle Q_{j,R}\left(x,t\right)Q_{i,R}\left(x^{\prime },t^{\prime }\right)\rangle =2k_{B}L_{ij}^{A}\delta \left(x-x^{\prime }\right)\delta \left(t-t^{\prime }\right) \end{array}$$

In Equations (70) and (71) i,j are either q,V or *D*. The fluctuation-dissipation relations can be derived from Equation (68) by integrating over the cross section.

## 11. Discussion and Conclusions

We have presented a way to define the representative elementary volume (REV) of a porous medium of mixed porosity in terms of coarse-grained variables. For this set of variables, we have written the Gibbs equation for the REV, building on earlier work for the microscale [20]. We have argued that the additive nature of the contributions of the various phases, surface areas, contact lines and configurational contributions, make it possible to use Gibbs equation for the REV. Another argument rests on the weakly coupled nature of the relevant partition functions.

Using the Gibbs equation and the balance laws, we next derived the entropy production for transport of heat and mass in porous media, following the standard procedure in non-equilibrium thermodynamics for heterogeneous systems [3,4,5]. Doing this, we have assumed that the ergodic hypothesis applies. Experimental and numerical support is available, but only for a few network simulations [7,30] and Hele–Shaw cells with glass beads [31].

On the basis provided by non-equilibrium thermodynamics, we formulated fluctuation dissipation theorems [20]. The theorems apply for a set of coarse-grained variables for the REV which obey the Gibbs equation on the macro-scale. The theorems were specified for immiscible, non-isothermal, two-phase flow in a non-deformable medium. These formulations of the fluctuation-dissipation theorems for porous media are new. We are thus not able to refer to supporting experimental work at all. In principle, it should be possible to determine the random fluctuations that we describe, via optical techniques on a Hele–Shaw cell. A very first support for these ideas has, however, been obtained from network simulations. Winkler et al. [7] simulated self-correlation and cross-correlation coefficients in a network with two-phase flow in a honeycomb lattice. The pore flow was modeled with the Washburn equation. The authors found that the matrix of coefficients obtained from auto and cross correlations of the fluctuating component flows was symmetric. This is the first result that indicates that Onsager reciprocal relations are obeyed for flow in a porous medium of coarse-grained variables.

The fluctuating flux–flux correlation functions in their study decayed fast on a macroscopic time scale (around 10${}^{-3}$ s). In this paper we have shown that their integrals resulted in $2k_{B}$ times the Onsager conductivities $L_{ij}^{P}$ where i,j are either *w* or *n*. More systematic studies of this sort would be very instructive for the next steps to be taken. A crucial test will be to obtain the Onsager relations from the fluctuation-dissipation theorem, as well as from experiments defined directly from the flux-force relationships.

The $L_{ij}$-coefficients used here are local coefficients. This means that they apply to a particular REV, and they are functions of the REV state variables that were defined earlier. According to Onsager [34,36], they are not functions of the driving forces. When the functions are known, it becomes possible to integrate the flux equations over a series of REVs. We may then arrive in a situation where the flux becomes a non-linear function of the overall driving force. This property is not an argument against using relations which are linear on a local level, however.

Other coarse-graining procedures have been used to define the REV [32]. McClure et al. [28] and Khanamiri et al. [27] proposed to use a geometric state function for two-fluid flow in porous media based on Minkowski functionals, in order to characterize the complex arrangements of fluids and solid phases within a porous medium. The functionals were presented as separate, independent variables. Central variables are the volume, surface area and surface curvatures. To linear order, their variable construction seems similar to the one that follows from Hill’s method [21,22].

Validity of Gibbs’ equation for the course grained variables presented here was essential, as a basis for all following derivations. In a different approach of McClure et al. [9], Gibbs’ equation was formulated for the fluctuating variables.

The question has been raised in the literature about the origin of the energy dissipation as heat. In the present formulation, the dissipation is uniquely defined by the temperature of the surroundings, $T_{0}$, times the entropy production [5]. The sources of the dissipation are then the product pairs of conjugate driving forces and fluxes. Each pair contributes to the dissipation. A typical example is the replacement of one fluid by another (Haines jumps). The shift induces fluctuations, adding to the cross-correlation in Equation (65).

We have discussed that experiments and simulations are needed to test the proposed relations for consistency and performance. The advantage of constitutive equations presented here is that they can be used to determine permeabilities by experiments. They can also be obtained using the fluctuation-dissipation theorems for the flows. The two determinations should give the same result for the same overall driving force (pressure difference). Coefficients are often determined at steady state conditions. The equations do not depend on there being a steady state, however. Once the coefficients are known, they could be used to model flow evolution in time on the time scale that is appropriate for the fluctuating flows.

## Data Availability

Not applicable.

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
