# Peer review of "Fluctuation-Dissipation Theorems for Multiphase Flow in Porous Media"

_entropy, 2021, doi:10.3390/e24010046_

Round 1
Reviewer 1 Report
Reviewer report on ENTROPY manuscript ID:entropy-1478619
TITLE: Fluctuation-dissipation theorems for transport through porous media
AUTHORS: Dick Bedeaux and Signe Kjelstrup
This manuscript extends Hill's ideas to multiphase flow in porous media.
The key concept of representative elementary volume allows to merge these
ideas to the standard non-equilibrium phenomenology developped by de Groot
and Mazur and other authors. This is done through Eq.(26) which had the
Gibbs equation structure. The authors apply these formal developments to
phase flow in Section 10.
Since the paper cautiously stays away from practical applications, it is not
possible to judge the extend of validity of this type of approach in an
realistic application.
Few points could be questioned, such as the assertion that the REV hypothesis
implies Gaussian white noise of random contributions (p10), or the validity
of the ergodic hypothesis in the context of a coarse graining in the REV
variables. These two points are not fully justified, and not intuitively
obvious either. The authors may wish to comment on them in order to clarify
their own ideas.
The paper is well written in the traditional form devoted to this type of
phenomenological approach, and is suitable for publication in Entropy in its
present form.
Reviewer 2 Report
The paper addresses the thermodynamic analysis of porous media. To this end authors extended the idea of the representative elementary volume. Using the non-equilibrium thermodynamics approach they obtain the entropy production and related heat and mass transport in porous media. Some fluctuation-dissipation theorems were established.
The paper is well-written, the results sound good. I may propose to discuss a bit more the additive nature of various contributions. In particular it could be interesting to discuss the case of changing porosity and/or presence of surface tension. For saturated porous media surface tension may play a crucial role. In addition, I may propose to discussion of some new techniques of RVE such as https://doi.org/10.1115/1.4043475, where a fast statistical homogenization procedure was discussed.
